# Using Adaptive Logics for Expression of Context and Interoperability in DL Ontologies

**Thierry Louge** [1,*] , **Mohamed Hedi Karray** [2] and **Bernard Archimède** [2]

[1] Calcul en Midi-Pyrénées (UAR CALMIP 3667), Université de Toulouse, Centre National de la Recherche Scientifique (CNRS), Institut National Polytechnique Toulouse (INP), Institut National des Sciences Appliquées (INSA), Institut Supérieur de l'Aéronautique et de l'Espace (ISAE), Université Paul Sabatier (UPS), 31400 Toulouse, France

[2] Le Laboratoire Génie de Production de l'École Nationale d'Ingénieurs de Tarbes (LGP-INP-ENIT), Université de Toulouse, CEDEX, 65016 Tarbes, France; mkarray@enit.fr (M.H.K.); bernard.archimede@enit.fr (B.A.)

[*] Correspondence: thierry.louge@toulouse-inp.fr

**Abstract:** Ontologies are logical theories that are used in computer science for describing different items such as web services, agents in multi-agent systems, or domain knowledge. Many ontologies exist, expressing various domains of knowledge with different abstraction levels (domain ontologies, top-level ontologies, and task ontologies are the usual categories). The conceptualization of the knowledge contained in an ontology is subject to change, whether because the context of its use changes, because the domain evolves, or because an ontology needs to interoperate with other elements using other ontologies. Change in logical theories is a form of defeasible reasoning, in which some formulas need to be added or removed from a knowledge base. Adaptive Logics (AL) is a logic managing defeasible reasoning that we investigate in this paper for managing change in ontologies expressed with Description Logics (DL). The adaptation of AL for DL will help express the context in which formulas remain valid or can be added to a DL knowledge base, and ease the interoperability between ontologies.

**Keywords:** adaptive logics; ontologies; interoperability; description logics

## 1. Introduction

Ontologies are at the core of the Semantic Web (SW), which is an ongoing evolution of the existing web architecture proposed by Berneers-Lee et al. [1]. Ontologies provide a semantic representation of knowledge contained in documents and Web Services (WS), which are referred to as Semantic Web Services (SWS) in this context. A very-well-known definition of ontologies has been given by Gruber, following which "*An ontology is an explicit specification of a conceptualization*" [2]. Ontologies are representd by logical theories, very often by the means of a family of logics called Description Logics (DL) where the expressivity of a given DL depends on the constructors that it uses [3].

As an expression of a conceptualization, ontologies are subject to change (e.g., to include new elements from the domain that they conceptualize). They are also subject to change in order to provide interoperability with other ontologies, as one way of obtaining interoperabillity between two or more ontologies is to merge those ontologies into a new one. In such a case, the resulting ontology is an enrichment of one of the input ontologies with rules from the other input ontologies. Therefore, in this paper, we will discuss ontology interoperabillity as a special case of change in ontologies.

One of the most influential works concerning belief changes in logical theories is the work of Alchourròn, Gardenfors, and Makinson that formulates the AGM postulates [4], proposing three (3) operators (revision, contraction, expansion) together with the properties that a logic should satisfy to be AGM-compliant. Expansion is the simple addition of a

formula to a given theory with which it is consistent. Revision, on the other hand, is the addition of a formula that is inconsistent with the theory, providing that the resulting theory is consistent. Finally, contraction is the operation by which a logical formula is rejected from a theory. The three operators defined in [4] ensure that the resulting theory is both consistent and closed under logical consequences.

It is well-known that the AGM postulates do not apply easily to DL, and several works exist for solving the problem of revision in DL. In a 2008 survey [5], five (5) approaches for revision in DL are compared following seven (7) criteria among which the *implementation criteria* stating whether there is an algorithm for the criteria or not and whether or not the algorithm has been implemented. Only one (1) approach had an algorithm available. Many recent works on change in ontologies focus on AGM postulates adaptation, while only a few of them come with different approaches such as peer-to-peer dialogue between ontologies [6].

Expression of context is another concern when using ontologies, as some of the rules expressed in the knowledge base may be only valid under certain circumstances. This may be considered as a specific case of revision, indicating which are the conditions under which some logical formulas are true and under which they are false, implying a revision of the ontology. Nonetheless, unlike "regular" revision, a revision linked to a context does not lead to the exclusion of formulas from the knowedge base. Indeed, those formulas may become true again when the context changes and are only temporarily considered false. The expression of context for DL is possible by using two-dimensional DL language: one for the knowledge base and one for the expression of the context [7].

Defeasible reasoning in DL is an active research topic, and extensions for DL providing such capabilities are proposed [8], for which guideleines for a concrete use remain vague and may require a consequent effort of adaptation of existing knowledge bases.

The merging of ontologies and expression of context can be considered as special cases of defeasible reasoning that allow inferences to be concluded and possibly retracted with the arrival of new information (i.e., rules or facts) inside the logical theory. Information about the context may be the kind of information susceptible to cause the retractation of some inferences and the conclusion of new ones. This is also the case when an ontology is enriched with the content of another one such as in ontology merging.

In this paper, we propose an approach for enhancing interoperability between OWL ontologies and the expression of context that relies on defeasible reasoning by the use of Adaptive Logics (AL). AL has been described by Diderik Batens [9], and a further description focusing on the standard format for AL for modeling defeasible reasoning has been given by Christian Straßer [10]. We will use the classical definition of inconsistency: a theory is inconsistent if it contains both a statement $r$ and its negation $\neg r$. By the means of AL, our work aims at ensuring the consistency of an ontology during its merging and the expression of its context. This paper presents an approach with implemented algorithms in order to provide expression of context and paraconsistency during merging for OWL ontologies.

The tags used for identifying abnormalities and context will be directly integrated into OWL and manipulated through simple algorithms. The approach and algorithms should also remain valid for every DL used, regardless of its expressivity. We propose to use the AL framework to accomplish this goal, as it is a well-studied, logically sound, and solid framework. Nevertheless, there has been no initiative in our knowledge for the integration of AL in DL neither for OWL nor any other ontological language.

Our contributions in this attempt for using the AL framework with DL are as follows:

1.    A set of XML tags identifying abnormalities and context;
2.    A set of algorithms for manipulating the tags and applying the adaptive proofs to OWL ontologies;
3.    A validation from different use-cases of ontology contextualization.

In Section 2 of this paper, we will give an overview of the ongoing works on the detection and correction of inconsistencies in ontologies, the expression of context, and the adaptive logics. The motivations of this paper will be detailed. Section 3 will expose

the elements that we propose as additions to OWL for expressing context, inconsistencies correction, and the algorithms for using those elements. The experiments conducted on test ontologies will be presented in Section 4, and Section 5 will conclude the paper with final remarks and future works.

## 2. Related Work

### 2.1. Preliminaries

One of the difficulties one encounters when working with ontologies is to define what is an ontology. This problem has been deeply discussed by Neuhaus [11], who investigates the incidences of a variant of the definition we gave in the introduction as an ontology being "an explicit specification of a formalization" [2]. As Neuhaus shows in his paper, both technical and philosophical points can be argued against this definition, and it is not in the scope of this paper to discuss whether or not, or to which extent we agree or disagree with Neuhaus. We only state that his work points out that to provide a clear, understandable, and working definition of an ontology is still a problem to be solved. In this paper, we will consider an ontology as a logical theory axiomatizing some fixed domain of discourse that is of interest for the ontologist. This axiomatization may be incomplete regarding the domain of discourse, but it needs to be efficient and complete for the ontologist. A more formal definition is given in Section 3.1.

### 2.2. Dealing with Change and Inconsistencies

Ontologies are not tied to a specific language or a specific logic. Very often, the logic used for ontologies is one of the multiple DL, and the language is OWL or OWL2, but this is not mandatory. Dealing with inconsistencies when there is a change in ontologies is a difficult task and a subject of research, as many different DLs exist that differ from the constructors that they use.

AGM postulates do not directly apply to DL. In their work for generalizing AGM postulates so that they become applicable to DL and other non-classical logics, Aiguier et al. [12] use relaxations to define revision operators satisfying weakened AGM postulates. The resulting theoretical framework is demonstrated with two DL: $\mathcal{EL}$ and $\mathcal{ELU}$. Relaxations and operators need to be redesigned following each DL specificity, which is not an easy task. Moreover, while $\mathcal{EL}$ and $\mathcal{ELU}$ are investigated in the paper, it is uncertain whether the framework applies for every DL and would need to be proven for each DL. Finally, there is no algorithm or concrete implementation of the framework that emphasizes the theoretical aspects of belief revision.

Another extensive work on changes in ontolgogies (more specifically, in the network of ontologies) has been conducted by Jérôme Euzenat [13]. More specifically, in his work, Euzenat focuses on the network of ontologies, which is a special case of interoperability where all ontologies can be used in combination with consistency. The network is considered as a global entity containing all the ontologies that are part of the network with their associated alignments. The network is normalized, which means that it contains one and only one alignment between each pair of ontologies. The definition of consistency used by Euzenat is that a network of ontologies is consistent if it has a model (i.e., if there exists a set of alignments and a set of ontologies for whom every assertion for every ontology is satisfied). The revisions considered by Euzenat may occur at two (2) levels: change in alignments or change in ontologies, and some postulates for handling revision in alignments are proposed in the same way that AGM postulates are expressed. The operators proposed by Euzenat for belief revision in the network of ontologies may need to be refined following the DL that is used in the ontologies of the network, and no concrete implementation of those operators for practical revision is proposed.

In their work concerning changes in ontologies, Dhelim et al. [14] propose a Spatial–Temporal–Logical Framework (STLF) focusing on representing temporal changes in spatial relations between physical objects. Then, the changes are represented by the means of time interval definitions (temporal parts) where properties apply. A change in the properties

is a change in the time interval for a considered object and vice versa. The composed relations linking spatial and temporal properties are expressed through SWRL rules. This approach seems promising for expressing physical properties depending on time, and it may be applicable for a more generic change representation management by replacing the expression of time intervals by the expression of context as conditions for change. However, ensuring the changes cause no inconsistency in the ontology is not in the scope of STLF, contrary to AL that focuses on defeasible reasoning rather than change management. Writing SWRL rules that may reproduce the AL functioning would be very difficult if even possible mainly because negation and modification (including rectraction) are not supported by SWRL. Moreover, the resulting set of rules would be even more difficult to understand than to write.

Peer-to-peer inference systems (P2PIS) is a field in which interoperability between agents is considered, each agent exhibits its own logical theory. Each peer theory being consistent and immutable, the interoperabillity between agents is ensured through mappings between the different theories. Therefore, those mappings constitute the only source of inconsistencies in such a case. As there is no overall knowledge of the system but only local theories with local mappings with other peers, an incoherence resulting from a mapping is stored by the peer who identifies it. The identification itself is based on a recursive query to all neighbors [15]. A similar algorithmic approach by the means of a negotiation dialogue between agents is proposed in Jiménez-Ruiz et al. [6]. The negotiation is conducted with different messages (e.g., *assert, repair* and *close*) together with weights acting as trust values given by an agent for a correspondence. Agents agree on either accepting or rejecting a correspondence after negotiations stating whether or not the correspondence is rejected, repaired (i.e, other correspondences need to be removed to maintain consistency if the ongoing correspondence is accepted) or accepted. Both [6,15] are purely algorithmic. In [15], the P2PIS is interpreted with propositional logics, whether the work in [6], namely addresses ontologies without further specifying which logic the ontologies are expressed with.

Haase and Stojanovic [16] proposed a method for dealing with inconsistencies in changes in ontologies expressed in OWL DL language, with underlying $\mathcal{SHOIN}$ DL. They separate three (3) different kinds of inconsistencies: structural, logical, and user-defined. The structural consistency ensures that only constructs supported by OWL DL are used in the axioms by re-expressing some of them when needed. When a re-expression is not possible, an approximation or even a removal may be needed. The logical consistency is expressed as "an ontology $\mathcal{O}$ is consistent if $\mathcal{O}$ has a model", which is a widely used definition of consistency in ontologies. Frequently, many different solutions can be found in order to bring back consistency, and some action from the user is needed. When this is the case, the user should specify what are the changes to be made in the ontology that makes the approach semi-automatic. Finally, user-defined consistency is not expressed in the ontology itself but rather in the user requirements (e.g., quality of modeling). The method from Haase and Stojanovic is implemented in KAON2 (http://kaon2.semanticweb.org/, accessed on 31 January 2022), which is an ontology management system.

### 2.3. Expression of Context in Ontologies

As an axiomatization of a domain of discourse, an ontology is not directed toward a specific use-case or a specific situation. As a consequence, it is sometimes necessary to find ways for adapting the use of an ontology in a given context. However, what is a context, and how can the use of an ontology be adapted to fit in a given context?

An example of using the term "context" in ontologies in a clinical situation is given by Bouamrane, Rector, and Hurrel in [17]. By using modular OWL ontologies, a comprehensive view of a personal medical context for a patient is provided. Following this context, reasoning is conducted, and recommendations are made for this patient's preoperative tests. In this case, the context designates the application of the same set of rules following a specific set of conditions rather than a change in the set of rules that apply for the reasoning.

It is quite a different definition of "context" than the definition we use in this paper and that we give at the end of the current section.

An in-depth investigation of the definition of context has been conducted by Bazire and Brézillon [18], based on one hundred and fifty (150) definitions of "context" coming from different domains, which include computer science. As a conclusion for their work, Bazire and Brézillon state that there is no consensus for whether or not a context should be static or dynamic, internal or external, or on the nature of the context itself (e.g., a set of information or a process). Nevertheless, a generic definition is proposed as "*The context acts like a set of constraints that influence the behavior of a system (a user or a computer) embedded in a given task*" [18]. A model for representing context is also given that separates the environment from the context. The main point of this model is that it emphasizes on the moment when a task is conducted and the goal of the task, as for Bazire and Brézillon, "*the reason why definitions diverge is that they don't put their focus of attention on the same topic*". So, the topic about which reasoning is conducted may imply a change in the rules that are used for the reasoning.

Different definitions of context are found in the literacy; for Mitra et al., the definition is the following: "*We consider context as any information that assists in determining users' QoE [Quality of Experience]*" [19]. In their work, Mitra et al. consider the context as an expression of the environment, as "*Dynamic context may include user location, velocity, network load, battery power, memory/CPU utilization, presence and signal to noise ratio*" [19]. Their approach uses the Context Spaces Model (CSM) from Padovitz et al. [20], relying on sensor readings and heuristics to identify the context (e.g., the user is in the meeting room or is attending a presentation).

A method for dealing with context specifically in DL has been proposed by Klarman and Gutiérrez-Basulto [7], where contexts may be described, organized, and manipulated as formal objects in first-order logic statements. In order to explicitly manipulate contexts separately from the ontologies that they contextualize, Klarman and Gutiérrez-Basulto present a description logics of context for manipulating contexts, which helps solving semantic interoperability problems of ontologies. The problems addressed by the two-dimensional DL proposed in [7] are those of concepts alignments, semantic importing, and the definition of axioms that act as upper-ontologies.

In a similar way that Klarman and Gutiérrez-Basulto define a language for contexts, Barkat et al. [21] separate the expression of context from the ontologies using a specific context language, which is formalized using a UML class diagram. This model makes use of external ontologies to express some elements of a context, such as temporal elements or units of measure. The work of Barkat et al. aims at an automatic construction of data warehouses from a set of user requirements. The expression of context act as a filter from the requirements to the concepts in the ontologies stored in a semantic database containing multiple ontologies. The requirements are mapped with the elements contained in the expression of context, and the concepts and instances resulting from those mappings are extracted from the source ontologies through a set of transformations. Those concepts and instances form the final data warehouse that is proposed to the user. Barkat et al. use an external model for expressing context and do not address the problem of logical consistency of the resulting theory. Their approach matches the context with the requirements, extracts the relevant instances and concepts from the available sources, and the final result is provided *as-is*.

The expression of context in ontologies is subject of ongoing research, and the work in this paper argues that this problem can be seen as a special case of defeasible reasoning and is therefore closely related with non-monotonic logics. Our point is that when a context is expressed a a logical rule (i.e., this is the case that the context holds), then other rules may change value and become true or false. Adaptive Logics (AL), which are introduced in the next subsection, are a logical formalism ensuring non-monotonicity and paraconsistency of the logical theories that they express. AL ensures that a change in the theory keeps the theory consistent; this change may come from the expression of a context or for another

reason. When two ontologies are merged, changes are the basis on which the new theory is built as new rules are added to the resulting ontology. That is why we investigate in this paper the use of AL for ontologies interoperability through ontologies merging and also consider the expression of context by the same means.

### 2.4. Adaptive Logics

Adaptive Logics are a family of logics similar to DL. This subsection will present the most important parts of the standard format for AL, which are introduced by Diderik Batens [9] and further investigated by Christian Straßer [10]. We encourage the reader to refer to [9,10] for an extensive view of the format or to the non-monotonic logics sections of the Stanford Encyclopedia of Philosophy [22] for an overview. An extensive study of adaptive logics as a "*generic formal framework for defeasible reasoning*" [23] is available in a book written by Christian Straßer [23] entirely devoted to the subject.

The framework of AL models is defeasible reasoning based on a lower limit logic **LLL**, a set of abnormalities $\Omega$, and an adaptive strategy. Given a set of premises $\Gamma$, AL will ensure that $\Gamma$ is interpreted "as normally as possible" [10] following the strategy. The two main strategies that are discussed in AL are *reliability* and *minimum abnormality*. We will now introduce the main elements of AL.

The **LLL** is the underlying logics for all reasoning in AL. It should satisfy some requirements, which are that **LLL** must be reflexive, transitive, monotonic, and compact. In order to ease the reading of this paper, we will use the same notation used by Christian Straßer: "$Cn_L(\Gamma)$ *denote the set of L-consequences of the premise set $\Gamma$*" [23].

Using the framework of AL for DL (regardless of its expressivity) requires that the theory expressed with DL be closed under logical consequences. With **LLL** being a DL, this closure will ensure the following:

1.  $\Gamma \subseteq Cn_{\textbf{LLL}}(\Gamma)$ (Reflexivity)
2.  If $\Gamma' \subseteq Cn_{\textbf{LLL}}(\Gamma)$ then $Cn_{\textbf{LLL}}(\Gamma') \subseteq Cn_{\textbf{LLL}}(\Gamma)$ (Transitivity)

DL are compact and monotonic logics by construction, so providing that the theory that they express is closed under logical consequences, they can be used as **LLL** for AL.

The set of abnormalities $\Omega$ contains formulas, which are expressed in the **LLL**. The function of $\Omega$ is to define conditions under which rules cannot be inferred, such as the following.

Where $\nu \in \Omega$, if $\phi \vdash \psi \vee \nu$, then $\psi$ follows defeasibly from $\phi$ on the assumption that $\nu$ is false (or, equivalently, that $\neg \nu$ is true) [22].

Adaptive proofs is the process by which AL:

*   Adds premises to the theory
*   Infers rules at some conditions or unconditionnally
*   Retracts (*mark*) rules or reintegrates (*unmark*) rules into the theory

Each new rule coming to the theory, as a premise or as a reasoning result, is a new *step* in the proof. As formulas are added as premises or inferred during adaptive proofs, $\Omega$ will grow to reflect that some rules are inferred *at some conditions (assumed to be false)* under which they would not apply. Those conditions are abnormalities that go into $\Omega$. The AL format uses the notion of *Dab*-formulas and *Dab*-consequences to represent the classical disjunction of members of $\Omega$. For $\Delta \subseteq \Omega, Dab(\Delta)$ is the classical disjunction of the members of $\Delta$, also referred as the *Dab*-formulas derived at no condition at stage *s* of the proof. "*The minimal Dab-consequences $Dab(\Delta)$ derivable from a given premise set $\Gamma$ are all $Dab(\Delta)$ for which (i) $\Gamma \vdash_{LLL} Dab(\Delta)$ and (ii) there is no $\Delta' \subset \Delta$ such that $\Gamma \vdash_{LLL} Dab(\Delta')$. For a minimal Dab-consequence $Dab(\Delta)$ we know that in each LLL-model of $\Gamma$ at least one of the abnormalities in $\Delta$ is validated. Due to the minimality of $\Delta$ there is no $\Delta' \subset \Delta$ with the same property. Where $Dab(\Delta_1), Dab(\Delta_2) \ldots$ are the minimal Dab-consequences, the set of unreliable abnormalities is $U(\Gamma) = \Delta_1 \cup \Delta_2 \ldots$*" [10]. This set of abnormalities $U(\Gamma)$ is investigated by the strategy when it comes to finally derive rules from $\Gamma$.

The next important element of the AL framework that we need to introduce here is the strategy. The standard format exposes two different strategies, *reliability* and *minimum abnor-*

*mality*. They differ on the way they deal with abnormalities. They share the main schema, following which a rule for which abnormality is satisfied is *marked* and consequently does not hold. The same rule can be *unmarked* at a later stage, then marked again and so forth. The difference comes when a new rule that consists in a disjunction of abnormalities for other rules appears in the theory.

Consider $1 : (a, \alpha)$ and $2 : (a, \beta)$: two couples expressing the same rule $a$ with two different abnormalitites $\alpha$ and $\beta$. When a new rule $c = \alpha \vee \beta$ comes into the theory (c being derived with no abnormality condition or integrated in the theory as a premise), the strategies lead to different behaviors. The *minimal abnormality* will consider that either 1 or 2 holds but not both, and therefore, the assumption $a$ still holds. The *reliability* strategy considers that any rule involving an abnormality that is part of a minimal disjunction in a rule derived at no conditions is marked. Therefore, 1 and 2 will be marked.

## 3. Expressing Context and Dealing with Inconstistencies Using Adaptive Logics

### 3.1. Expressing AL Elements in DL: Adaptive Context Expression (ACE)

In the AL standard format, $\Omega$ is enriched by reasoning on the formulas presented in the theory. New formulas may appear in the theory from its logical closure and create new entries in $\Omega$ specifying under which conditions those new formulas would fulfill original $\Omega$ content: The abnormalities for the rules are derived from the set of abnormalities $\Omega$. To introduce this behavior into the algorithms, we propose to specify an optional set of abnormalities per formula, in addition to the original $\Omega$ content, which will remain untouched by the algorithms. The content of $\Omega$ and every formula-related abnormality should be tested every time that a new formula comes in the theory. This allows to detect when a new formula causes an abnormality, whether by itself or by implying any other logical consequence that would cause an abnormality.

The coding of the algorithms will traduce abnormalities for the rules and for $\Omega$ into SARQL queries, with a simple mechanism: If the number of results from the query is not 0 (zero), then the abnormality is satisfied.

This expression of AL elements at two different levels—the level of the formulas and the general level of $\Omega$—allows the detection of inconsistencies in ontologies merging. Those inconsistencies may be defined in $\Omega$ such as, for example, the rule $\Omega = r \wedge \neg r$ or targeting a specific formula, that may become invalid in the theory when the abnormality (e.g., the expression of a change in the context) is satisfied.

This expression of a context is a special case of defeasible reasoning ensured by our approach. If a set of abnormalities is provided (e.g., $\Omega = r \wedge \neg r$), then the resulting ontology can not satisfy any element of $\Omega$ and will reject (*mark*) the formula leading to such abnormality. Some formulas may consequently be excluded from the ontology so that the consistency, as expressed in $\Omega$ and at the formula-specific level, is maintained. The number of ontologies that are merged is of no importance, as long as the abnormalities are expressed. Nevertheless, the alignments that may be necessary between the different ontologies need to be done before the merging.

### 3.2. Reasoning on ACE Elements (RACE)

This section presents the algorithms for reasoning on an ontology $\Gamma$ and AL expression that we propose.

The outline of the algorithms is as follows. The integration of a new rule in $A$ can have two effects:

1. It can cause an inconsistency in $A$. If that is the case, some rules in $A$ must be marked and go into $\varepsilon$. As a consequence;
2. Some rules in $E$ may become valid regarding their abnormalities toward the rules in $A$ now that some rule in $A$ has been marked. If this is the case, then the said rules should go from $\varepsilon$ to $A$ and another run of verifying if the new rule in $A$ may cause inconsistency is necessary, and so on until no more rules are added in $A$.

Here are some elements to help understand the algorithms:

$(a, \lambda)$ is a couple composed of a rule $a$ and the abnormalities $\lambda$ on $a$.

$\Gamma$ is the set of rules $(a, \lambda)$ composing the ontology that is under study.

$A$ is the set of $(a, \lambda)$ that is unmarked.

$E$ is the set of marked rules $(a, \lambda)$.

$\Omega$ is the set of abnormailites $o$ that apply for every rule $a \in \Gamma$.

In the next section, we present the tags for expressing abnormalities in ontologies expressed in RDF/XML syntax.

## 4. Expressing ACE Abnormalities

We propose a set of RDF/XML tags for expressing the abnormalities in the ontology, which are either associated with rules $(a, \lambda)$ or applied to the entire ontology ($\Omega$). These tags will be used in Section 5 in the experiments conducted to validate the algorithms and the effectiveness of our approach.

### 4.1. Expressing an Abormality on a Specific Rule

We use the attribute "**ace:Condition**" inside the tag specifying an ontology element to indicate that this element is bound to an abnormality.

Stating that including a named individual "Magnitude", the individual of the Concept "SeismCharacteritics", in the ontology should be done regarding a given abnormality named "abnormality1", is expressed as follows:

```
<owl:NamedIndividual
ace:Condition=''abnormality1'' rdf:about='Magnitude'>
<rdf:type
rdf:resource=''SeismCharacteristics''>
</owl:NamedIndividual>
```

The attribute "**ace:Condition**" used above references an element "ace:Condition", whose element "name" is "abnormality1". This element has an attribute named "**ace:Query**" which itself has two attributes, "**ace:negation**" and "**ace:abnormality**".

The element "**ace:Query**" indicates the logical rule defining the abnormality. In the following examples, $a : C$ denotes the membership of individual $a$ to concept $C$.

Let us consider the abnormality: $\neg(Geophysics : Context)$, stating that the abnormality is satisfied (and thus, that the formula must be marked) when the individual "Geophysics" is not a member of the concept "Context". The abnormality is a negation; we use the attribute "ace:negation" with the value "negative" to express this negation (otherwise, we would have indicated "no negation" as the value for "ace:negation").

Then, the abnormality is defined as follows:

```
<ace:Condition ace:name=''abnormality1''>
<ace:Query ace:negation=''negative'' ace:abnormality=
''Geophysics,rdf:type,Context;''/> </ace:Condition>
```

### 4.2. Expressing Conjunctions and Disjunctions in Abnormalities

An abnormality is not necessarily as simple as $\neg(Geophysics : Context)$. Let us consider the following abnormality: $a : C \wedge \neg(b : D)$, stating that the abnormality is satisfied when the individual $a$ is member of the concept $C$ AND the individual $b$ is not a member of the concept $D$. We need to express the conjunction and the negation of a part of the formula. The conjunction is expressed by separating the elements from the formula with ";" and the negation by "!" preceding the negated element.

$a : C \wedge \neg(b : D)$ becomes:

```
<ace:Query ace:negation="no negation"
ace:abnormality= "a,rdf:Type,C;!b,rdf:Type,D;"/>
```

A disjunction in the abnormality is expressed by separating the disjunction into different declarations. Let us consider the abnormality: $a : C \vee a : D$ stating that the abnormality is satisfied when the element a is a member of the concept C OR D.

The expression of this abnormality goes through two steps, defining an abnormality **with the same name** for each member of the general abnormality.

$a : C \vee a : D$ becomes:

```
<ace:Condition ace:name=''example2''>
<ace:Query ace:negation=''no negation''
ace:abnormality= ''a,rdf:Type,C;''/>
</ace:Condition>
<ace:Condition ace:name=''example2">
<ace:Query ace:negation=''no negation''
ace:abnormality= ''a,rdf:Type,D;''/>
</ace:Condition>
```

*4.3. Expressing the Set of Abnormalities* $\Omega$

The set $\Omega$ defines conditions that lead to the marking of the formula that causes their verification. The algorithms in Section 3 indicate when this set of abnormalities is checked and with what consequences. We indicate here the ace tag for specifying omega, which is simply named "ace:Omega". The formulas composing $\Omega$ follow the same expression as the formulas defining other abnormalities.

Let us consider $\Omega = \{g : C \wedge \neg(a : C)\}$, it becomes: `<ace:Omega>`

```
<ace:Query ace:negation=''no negation''
ace:abnormality=
''g,rdf:type,C;
!a,rdf:type,C;''/>
</ace:Omega>
```

In the following section, we use these expressions to express various contexts and their consequences on the formulas integrated in an ontology.

## 5. Experiments

*5.1. Preliminaries*

The code for re-running the experiments described in this section and the different version of the ontology that is used to run those tests are available on GitHub (https://github.com/ThiLou2/ACE, accessed on 31 January 2022).

The code is run as follows:

`python3 ACE_Verification.py ACEn.owl` for testing the version "n" of the ontology (the different versions of the ontology with their correponding elements of testing are described in Section 5.2).

The tests produce:

- An output ontology ACEn.owl_AceVerified.owl (when testing ontology ACEn.owl);
- A log file named ace.log.

*5.2. Tests Description and Results*

We conducted eleven (11) tests for validating our approach of context expression based on the minimal abnormality strategy of Adaptive Logics. We use an individual named "magnitude", which should be considered as a member of two different concepts: SeismCharacteristic and StellarParameters following the abnormalities defined in an ontology.

We have a concept named "Context" and two individuals that may be members of this concept: astrophysics and geophysics. Depending on the context (astrophysics, geophysics, none, or both), we want the formulas stating that ($magnitude : SeismCharacteristics$) or ($magnitude : StellarParameters$) be rejected or included in the resulting ontology following their abnormalities and $\Omega$.

The source ontology ACEn.owl is the $\Gamma$ ontology for Algorithm 1. It contains the elements with applicable abnormalities. The destination ontology ACEn.owl_AceVerified.owl is the $A$ returned by Algorithm 1. It contains the formulas that are unmarked following Algorithm 3 and does not contains the formulas marked following Algorithm 2 after run-

ning the abnormality strategy algorithm, and it is the resulting ontology containing all formulas from Γ satisfying this strategy.

It is important to mention that while for the sake of simplicity and code clarity, we gathered all formulas and abnormalities in a single source ontology, this is not a limitation. Multiple source ontologies may be read with the same algorithms and provide the same results as long as they share the concepts, individuals, and abnormalities URIs.

In the following, we present the tests conducted with the algorithms and ace tags.

---

**Algorithm 1:** RACE minimal abnormality strategy

---

**Ensure:** The resulting ontology is consistant

  1: **return** $A = \{(a, \lambda) \in \Gamma \backslash \Gamma \vdash_{AL} a\}$
  2: $A = \varnothing$
  3: $E = \varnothing$
  4: **for all** $(a, \lambda) \in \Gamma$ **do**
  5:   **if** $A \vdash_{LLL} \lambda$ **then**
  6:     $E = E + (a, \Lambda)$
  7:     continue
  8:   **end if**
  9:   **if** $A + a \nvdash o, \forall o \in \Omega$ **then**
10:     $A = A + (a, \lambda)$
11:     newrule = 1
12:   **else**
13:     $E = E + (a, \lambda)$
14:   **end if**
15:   **while** newrule = 1 **do**
16:     call RulesMarking
17:     call RulesUnmarking
18:   **end while**
19: **end for**

---

**Algorithm 2:** RulesMarking algorithm

---

**Ensure:** After the integration of a new rule in $A$, every $(a, \Lambda) \in A$ remains valid regarding its set of abnormalities $\Lambda$.

  1: newmark = 1
  2: **while** newmark = 1 **do**
  3:   mark = 0
  4:   **for all** $(a, \lambda) \in A$ **do**
  5:     **if** $A \vdash_{LLL} \lambda$ **then**
  6:       $E = E + (a, \lambda)$
  7:       $A = A - (a, \lambda)$
  8:       mark = 1
  9:       call RulesUnmarking
10:     **end if**
11:   **end for**
12:   *newmark ← mark*
13: **end while**

---

**Algorithm 3:** RulesUnmarking Algorithm

---

    **Ensure:** After the integration of a new rule in $A$, every $(a, \Lambda) \in E$ should remain marked regarding its set of abnormalities $\Lambda$.

1:  newrule = 0
2:  **for all** $(a, \lambda) \in E$ **do**
3:    **if** $A + a \nvdash o, \forall o \in \Omega$ **then**
4:      **if** $A \vdash_{LLL} \lambda$ **then**
5:        continue
6:      **else**
7:        $E = E - (a, \lambda)$
          $A = A + (a, \lambda)$
          newrule = 1
          break
8:      **end if**
9:    **else**
10:      continue
11:    **end if**
12: **end for**

---

### 5.3. Basic Tests

In ACE1.owl, we specify that magnitude is a SeismCharacteristic, with an abnormality stating that this is not the case when geophysics is not a context.

Using the terminology defined in Section 3, we have:
$(a, \lambda) = (magnitude : SeismCharacteristics,$
$\neg(geophysics : Context)$

In addition, no formula indicates in ACE1.owl that geophysics is a context, so $(geophysics : Context)$ is false, $\neg(geophysics : Context)$ is true and the abnormality is satisfied. As a result, after the test, the resulting ontology

ACE1.owl_AceVerified.owl does not includes the formula
*magnitude : SeismCharacteristisc*. Indeed, this formula is bound to an abnormality that is satisfied after running Algorithm 1, and the formula is consequently marked.

In ACE2.owl, the situation is the same, but we have a formula stating that $(geophysics : Context)$ is true. After the test, the resulting ontology does include the formula *magnitude : SeismCharacteristics*.

ACE3.owl begins testing the algorithms with multiple formulas and abnormalities. ACE3.owl states that magnitude is a SeismCharacteristic. The abnormality is that geophysics is not a context, as in the previous tests. ACE3.owl states that magnitude is a StellarParameters, the abnormality being that astrophysics is not a context. In ACE3.owl, we have *geophysics : Context* and $\neg(astrophysics : Context)$. After the test, the resulting ontology includes the formula $(magnitude : SeismCharacteristics)$, and it does not include the formula $(magnitude : StellarParameters)$.

### 5.4. Testing $\Omega$

As the $\Omega$ set of abnormailities applies to the integration of any formula into the ontology, it is necessary to properly test that it is taken into account and properly induces the marking of relevant formulas.

In ACE4.owl, $\Omega = (geophysics : Context \land astrophysics : Context)$. Omega is satisfied when geophysics and astrophysics are context altogether. In ACE4.owl, we first found that geophysics is a context, which does not satisfies any abnormality. Then, we encounter that astrophysics is a context.

According to Algorithm 1, as there is no condition for this formula, then at first attempt, it is included in the ontology. Then, omega is checked. It is found that omega is satisfied and the formula is marked. Therefore, when the formula indicating that magnitude is

a stellar parameter is tested, it is also marked as intended, because its abnormality is $\neg(astrophysics : Context)$. This allows us to see not only that $\Omega$ is taken into account, but also that the order in which the formulas appear in the knowledge base is not neutral, which wil be investigated in Section 5.5.

*5.5. Advanced Testing: Order of Appearance of Formulas in the Knowledge Base, Marking and Unmarking, Conjunctions and Disjunctions*

In this section, we investigate the influence of the order of appearance of the formulas in the knowledge base. We also illustrate the use of conjunctions and disjunctions and show a concrete functioning of the formulas marking/unmarking process.

In ACE5.owl, the first formula encountered is $(magnitude : SeismCharacteristics)$, and $(geophysics : Context)$ is encountered afterwards. The marking/unmarking process is the following (this can be checked by looking at the log file produces during the tests): When the formula $(magnitude : SeismCharacteristics)$ is encountered and tested, it is marked because $(geophysics : Context)$ is false, satisfying the abnormality for $(magnitude : SeismCharacteristics)$.

When $(geophysics : Context)$ is encountered and integrated in the knowledge base, by the marking/unmarking algorithms, invocation
$(magnitude : SeismCharacteristics)$ is unmarked and integrated in the knowledge base.

In ACE6.owl, we investigate what happens when stating that astrophysics is a context before stating that geophysics is a context. In this case, geophysics as a context is marked (because $\Omega = (geophysics : Context \land astrophysics : Context)$), and we keep $(astrophysics : Context)$ in the knowledge base. Finally, magnitude is a stellar parameter and not seismic characteristics, as indicated by the related abnormalities.

ACE7.owl can be used for testing a disjunction of conditions. In ACE7.owl, the formulas state that magnitude is a stellar parameter $(magnitude : StellarParameters)$ as long as astrophysics is a context and geophysics is a context, without specifying any $\Omega$ abnormality.

Then, the abnormality concerning $magnitude : StellarParameters$ follows:
$\neg(astrophysics : Context) \lor \neg(geophysics : Context)$, which is expressed through ace tags as:

```
<owl:NamedIndividual
ace:Condition=''abnormality2'' rdf:about=''Magnitude''>
<rdf:type rdf:resource=''StellarParameters''/>
</owl:NamedIndividual>
<ace:Condition ace:name=''abnormality2''>
<ace:Query ace:negation=''negative'' ace:abnormality=
''Astrophysics,rdf:type,Context;''/>
</ace:Condition>
<ace:Condition ace:name=''abnormality2''>
<ace:Query ace:negation=''negative''
ace:abnormality=
''Geophysics,rdf:type,Context;''/>
</ace:Condition>
```

In ACE7, it is also stated that astrophysics and geophysics are members of context. Therefore, the intended magnitude is a member of StellarParameters after the test.

ACE8 completes the test conducted with ACE7.owl but with no formula indicating that geophysics is a context. Therefore, the intended magnitude is not a member of StellarParameters after the test.

We used ACE9.owl to test a conjunction of conditions. In this version of ACE, magnitude is a stellar parameter as long as astrophysics is a context and geophysics is a context (abnormality:$\neg(astrophysics : Context) \lor \neg(geophysics : context)$), taking into account $\Omega = \{astophysics : Context \land geophysics : Context\}$. ACE9.owl includes formulas stating that both astrophysics and geophysics are contexts, which triggers the abnormality in $\Omega$. Therefore, the algorithms detect that $\Omega$ is verified and the formula expressing the second

context is marked (*geophysics* : *Context*). As a consequence, magnitude is not a member of StellarParameters after the tests, although astrophysics is still a context.

A last couple of tests concerning $\Omega$ complete our round of tests.

In ACE10, we have:

$\Omega = \{geophysics : Context \wedge \neg(astrophysics : Context)\}$.

As in ACE10.owl, both geophysics and astrophysics are members of Context, while *Omega* does not lead to any rule marking. On the contrary, ACE11.owl does not include the formula *astropysics* : *Context*. After testing, neither astrophysics nor geophysics are members of Context, and magnitude is neither a StellarParameters nor SeismCharacteristics.

## 6. Summary of the Tests

In this section, we summarize the tests described in Section 5 in order to provide a comprehensive view of the abnormalities tested in the different versions of ACEn.owl. The results are shown in Table 1, where the "Source" column indicates which formulas are embedded in the source ontology ACEn.owl. Columns $(a_1, \lambda_1)$ and $(a_2, \lambda_2)$ indicate the couples formula/abnormalities embedded in ACEn.owl. The "Unmarked" column shows the resulting formulas included in the ontology after the tests (unmarked). For keeping the table easy to read, we omit the excluded (marked) formulas.

Formulas in the table are:

- F1: *magnitude* : *SeismCharacteristics*
- F2: $\neg(geophysics : Context)$
- F3: *geophysics* : *Context*
- F4: *astrophysics* : *Context*
- F5: $\neg(astrophysics : Context)$
- F6: *magnitude* : *StellarParameters*.

**Table 1.** Summary of RACE tests.

| Tested in | $\Omega$ | $(a_1, \lambda_1)$ | $(a_2, \lambda_2)$ | Source | Unmarked |
|---|---|---|---|---|---|
| ACE1.owl | - | F1,F2 | - | - | - |
| ACE2.owl | - | F1,F2 | - | F3 | F1 |
| ACE3.owl | - | F1,F2 | F6,F5 | F3 F5 | F1 |
| ACE4.owl | F3∧F4 | F1,F2 | F6,F5 | F3 F4 | F1 |
| ACE5.owl | F3∧F4 | F1,F2 | F6,F5 | F3 F4 | F1 |
| ACE6.owl | F3∧F4 | F1,F2 | F6,F5 | F3 F4 | F6 |
| ACE7.owl | - | F6,F5∨F2 | - | F3 F4 | F6 |
| ACE8.owl | - | F6,F5∨F2 | - | F4 | - |
| ACE9.owl | F3∧F4 | F6,F5∨F2 | - | F4 F3 | F4 |
| ACE10.owl | F3∧F5 | F1,F2 | F6,F5 | F4 F3 | F6 F1 |
| ACE11.owl | F3∧F5 | F1,F2 | F6,F5 | F3 | - |

## 7. Conclusions and Future Works

In this paper, we presented algorithms and XML tags for expressing Adaptive Logics elements into ontologies expressed in DL. This elements allow expressing formulas with abnormalities, whose logical truth excludes the related formula from the ontology. It also expresses a set of general abnormalities ($\Omega$), toward which new formula candidates to integrate the ontology are tested. If the new formula makes the ontology verify $\Omega$, it is rejected from the ontology unless a new formula renders it compatible with the abnormalities, going through the marking/unmarking process. This is an example of handling defeasible reasoning in ontologies expressed with DL.

This work allows expressing the logical truth of formulas regarding the expression of a context, as shown in the tests.

More generally, it allows expressing under which conditions a formula should be rejected from an ontology for keeping the knowledge base sound. This will be helpful for interoperability between ontologies, to merge different ontologies into a single one, ensuring that no abnormality is kept in the resulting ontology.

This work has some limitations:

- Alignment between the ontologies to merge needs to be ensured before the merging.
- Complex abnormalities (e.g., $a \wedge (b \vee (\neg c \vee d))$) can be tricky to express.
- The order in which formulas, related to other formulas' abnormalities, are encountered somewhat matters. A formula, unbound to any abnormality and part of (more than one) other formula abnormality, may induce a precedence of one formula toward another. This is shown in the tests using ACE6.owl. Apart from this specific case, the order in which formulas and abnormalities are encountered does not matter.

Future works concern the adoption of the tags and algorithms presented in this paper for the interoperability and context expression of ontologies. We investigate the development of a plug-in for Protege editor to help this adoption.

AL has two main strategies for dealing with abnormalities: the minimal abnormality strategy and the maximum reliability strategy. The strategy presented in this paper is the former, while the development and testing of algorithms to ensure the latest is on its way.

**Author Contributions:** T.L.: Conceptualization, Formal analysis, Methodology, Writing original draft, Writing review and editing. M.H.K.: Conceptualization, Formal analysis, Methodology, Writing original draft, Writing review and editing. B.A.: Supervision, Conceptualization, Formal analysis, Writing review and editing. All authors have read and agreed to the published version of the manuscript.

**Funding:** This research was partially financed by the OntoCommons project funded by the European Union's Horizon 2020 research and innovation programme under Grant Agreement no. 958371.

**Institutional Review Board Statement:** Not applicable.

**Informed Consent Statement:** Not applicable.

**Data Availability Statement:** Not applicable.

**Conflicts of Interest:** The authors declare no conflict of interest.

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
