# Peer review of "Using Adaptive Logics for Expression of Context and Interoperability in DL Ontologies"

_information, doi:10.3390/info13030139_

Round 1

Reviewer 1 Report

this paper proposes an approach for enhancing interoperability between
OWL ontologies and expression of context that relies on defeasible reasoning by the use of adaptive logics (AL).

The contributions are clear and the paper is well-written and easy to follow. I have just few comments that may help improve the manuscirpt:

1-to highlight your contributions, move subsection 2.5 to the end of the introduction and merge it with the current contributions paragraph.

2-what is the advatages of using the proposed AL based approach over rule based reasoning such as SWRL rules.

3-in the experiment section, you did not validate your proposed method for detecting abnormality in ontologies, it would be better if you can compare your method with other approaches that does not use AL

4- optionally, you can add the following works as related works:
[1] Dhelim et al. "STLF: Spatial-temporal-logical knowledge representation and object mapping framework." 2016 IEEE International Conference on Systems, Man, and Cybernetics (SMC). IEEE, 2016.
[2] Bouamrane, Matt-Mouley, Alan Rector, and Martin Hurrell. "Using OWL ontologies for adaptive patient information modelling and preoperative clinical decision support." Knowledge and information systems 29.2 (2011): 405-418.

Author Response

The contributions are clear and the paper is well-written and easy to follow. I have just few comments that may help improve the manuscirpt:

We would very sincerely thank the reviewer for his or her attentive reading of our work and the remarks and suggestions. That helps to make the paper more sound and easy to follow.

1-to highlight your contributions, move subsection 2.5 to the end of the introduction and merge it with the current contributions paragraph.

We much thank the reviewer for this suggestion that we followed, and that helps the reading of the paper by making the purpose of this work much clearer from the beginning.

2-what is the advatages of using the proposed AL based approach over rule based reasoning such as SWRL rules.

In our mind, the advantage of using AL based approach is to rely on a theoreticaly sound framework that does not depend on the expressivity of the DL considrered. Providing algorithms for applying AL, based on xml tags, solves the problem of writing SWRL that would hardly reproduce the AL framework funtionning mainly because negation and modification (including rectraction) are not supported by SWRL.

3-in the experiment section, you did not validate your proposed method for detecting abnormality in ontologies, it would be better if you can compare your method with other approaches that does not use AL

The inner AL functioning takes the set of abnormalities into account, retracting or reintegrating in the ontology rules that combination of which lead to abnormalities. The tests described in section 5 and the summary of their results in section 6 refelcts how the abnormalities are detected and, consequently, which rules are left out or incorporated in the resulting ontology after running the AL algorithms. We agree that comparing our results with other approaches would have been interesting, and this is probably something we should do in the future. For this specific paper, we intent to provide a way to apply AL for defeasible reasoning and to show how rules are marked or unmarked, and thus how the context is taken into account.

Running the same set of tests following other approaches will need more time than we have for re-submitting this work and would fully fit, in our point of view, in a paper comparing the merits and flaws of the main approaches for expressing context in ontologies with consistency checks. Such a comparison should also take into account important criteria such as the expressivity of the different logic following the approaches considered. We fully agree that such a survey would be of great interest.

4- optionally, you can add the following works as related works:
[1] Dhelim et al. "STLF: Spatial-temporal-logical knowledge representation and object mapping framework." 2016 IEEE International Conference on Systems, Man, and Cybernetics (SMC). IEEE, 2016.
[2] Bouamrane, Matt-Mouley, Alan Rector, and Martin Hurrell. "Using OWL ontologies for adaptive patient information modelling and preoperative clinical decision support." Knowledge and information systems 29.2 (2011): 405-418.

We would sincerely like to thank the reviewer for pointing out these papers. We found them very interesting reads, and changes in section 2.2 for [1] and section 2.3 for [2] have been made in the text to reflect our thoughts and what we learned by those readings.

Reviewer 2 Report

The paper presents the usage of Adaptive Logics (AL) for expression of context and interoperability in Description Logics ontologies. According to the Authors, the adaptation of AL for DL helps expressing the context in which formulas remain valid or can be added to a DL knowledge base, and ease the interoperability between ontologies. The topic is interesting and the paper is well corresponding to the journal aim and scope.

The paper is well structured. The Authors presented the related work and their motivations and contributions. The Authors made the effort by providing the set of algorithms for manipulating the tags and applying the adaptive proofs to OWL ontologies. They tested their work in the section 5. It enabled to express the logical truth of formulas regarding the expression of a context.  

Overall, the article presents a very interesting approach to solving an important issue in the context of connecting ontologies and ensuring its interoperability. The mentioned development of the Protege plugin can definitely support the process described by the Authors.

Minor typos:

Table 1 refers to section 6 – it is better to place it before conclusions section.

The list of references should be extended. There are no new references.

Author Response

The paper presents the usage of Adaptive Logics (AL) for expression of context and interoperability in Description Logics ontologies. According to the Authors, the adaptation of AL for DL helps expressing the context in which formulas remain valid or can be added to a DL knowledge base, and ease the interoperability between ontologies. The topic is interesting and the paper is well corresponding to the journal aim and scope.

The paper is well structured. The Authors presented the related work and their motivations and contributions. The Authors made the effort by providing the set of algorithms for manipulating the tags and applying the adaptive proofs to OWL ontologies. They tested their work in the section 5. It enabled to express the logical truth of formulas regarding the expression of a context.  

Overall, the article presents a very interesting approach to solving an important issue in the context of connecting ontologies and ensuring its interoperability. The mentioned development of the Protege plugin can definitely support the process described by the Authors.

We would first like to deeply thank the reviewer for supporting our work. We believe that the text has been improved, thanks to the comments and also the comments from reviewer #1. 

Changes have been made in the text with the deletion of part 2.5 that has been included at the end of the introduction section.

In parts 2.2 and 2.3, two more papers have been considered.

Minor typos:

Table 1 refers to section 6 – it is better to place it before conclusions section.

Thanks for pointing this out, the table has been repositionned.

The list of references should be extended. There are no new references.

Thanks again for this important point. Three more references have been considered, in the sections 2.2 and 2.3 as stated aboce, and also a recent work ( P. A. Bonatti, I. M. Petrova, and L. Sauro, “Defeasible reasoning in de-scription logics: an overview on dlˆ n,”arXiv preprint arXiv:2009.04978,2020) in the introduction section on page 2.

Round 2

Reviewer 1 Report

the authors have addressed all my comments